# Automated Scheduling Approach under Smart Contract for Remote Wind Farms with Power-to-Gas Systems in Multiple Energy Markets

**Zhenya Ji, Zishan Guo \*, Hao Li and Qi Wang**

School of Electrical and Automation Engineering, Nanjing Normal University, No. 2 Xueyuan Road, Nanjing 210046, China; jizhenya@njnu.edu.cn (Z.J.); shadowix_lh@outlook.com (H.L.); wangqi@njnu.edu.cn (Q.W.)
\* Correspondence: zishanbang@126.com

**Abstract:** The promising power-to-gas (P2G) technology makes it possible for wind farms to absorb carbon and trade in multiple energy markets. Considering the remoteness of wind farms equipped with P2G systems and the isolation of different energy markets, the scheduling process may suffer from inefficient coordination and unstable information. An automated scheduling approach is thus proposed. Firstly, an automated scheduling framework enabled by smart contract is established for reliable coordination between wind farms and multiple energy markets. Considering the limited logic complexity and insufficient calculation of smart contracts, an off-chain procedure as a workaround is proposed to avoid complex on-chain solutions. Next, a non-linear model of the P2G system is developed to enhance the accuracy of scheduling results. The scheduling strategy takes into account not only the revenues from multiple energy trades, but also the penalties for violating contract items in smart contracts. Then, the implementation of smart contracts under a blockchain environment is presented with multiple participants, including voting in an agreed scheduling result as the plan. Finally, the case study is conducted in a typical two-stage scheduling process—i.e., day-ahead and real-time scheduling—and the results verify the efficiency of the proposed approach.

**Keywords:** integrated energy system; scheduling; energy trade; smart contract





## 1. Introduction

With the aggravation of global energy security and environmental pollution problem, various renewable energy types—especially the wind energy—have become the focus on large-scale development and utilization. Due to the limitation for local absorption of intermittent wind power in remote areas, the phenomenon of power curtailment exists in large quantities, which compromise the carbon reduction and economic benefits of wind farms [1]. The power-to-gas (P2G) technology, with its advantages of reducing renewable energy curtailment and consuming carbon, has become a necessary supplement for remote wind farms [2]. By taking P2G in the scheduling plans in multiple energy forms—which include electricity, gas, and carbon—it can further enhance the economic benefits, while improving the wind power accommodation and reducing carbon emissions [3].

The introduction of P2G systems makes the scheduling of power from wind farms not only a matter of the electricity market, but also encompasses gas market and carbon market. A typical scenario for wind farms with P2G embedded is that surplus power that is not sold in the electricity market due to bidding strategies, transmission constraints, etc., can be utilized through P2G [4], while also participating in the gas and the carbon markets. Many researchers have studied such kinds of optimal scheduling of wind farms with P2G embedded in multiple energy markets [5–7]. In [8], the market behaviors of power systems and natural gas systems which are coupled by P2G are described considering the influence of the market pricing mechanism on the coordinated optimal scheduling.

The concept of a combined P2G and gas-fired generator system is brought up in [9], and the optimal scheduling is studied considering renewable energy accommodation and the ability to reduce carbon emissions. The potential of P2G to absorb renewable energy is assessed in [10], and then the optimal scheduling results of the electricity and gas markets considering the impact of P2G on long/short term natural gas prices is analyzed. However, these studies described P2G with crude models. A P2G system has a high investment and operation cost [11], and the adoption of simplified P2G models would lead to inaccurate results. Improvements thus can be made to enhance the accuracy of scheduling results.

Moreover, most wind farms are geographically remote and spatially dispersed due to constraints in wind farm siting [12,13]. It leads to a decentralized form of information exchange, so cyber instabilities—such as delays, dropouts, and tampers, etc.—are hard to avoid and difficult to fix in time, even if they can be detected immediately. In this context, wind farms would face the issue of communication instability and insecurity. In addition, the electricity, gas, and carbon markets are generally managed by various organizations in different locations and are often with different temporal scales. When wind farms located in remote areas participate in multiple energy markets, the possibility of information dissonance increases. Any untimely or missing or falsified information with one market organizer can affect the scheduling results. Therefore, when the ideal stable and secure communication environment is not assumed, improvements need to be made regarding how to guarantee the effectiveness of scheduling plans in multiple energy markets [14].

To deal with the above-mentioned problems, an automated scheduling approach under blockchain-enabled smart contracts can provide an effective solution. Being a distributed database, the blockchain technology facilitates the prevention of information tampering, ensures the security of transactions, and provides the ability to automate the execution of transactions/settlements [15,16]. Meanwhile, smart contracts are able to execute pre-determined contracts automatically and securely. Smart contracts in the blockchain environment are thus able to automate the contract procedures and minimize interactions between market organizers [15,17].

Researchers have carried out exploratory studies and applications in related energy fields [14,18] and market transactions [19]. A pilot project for an energy system in Japan in [20] analyzes the multiple challenges to the expansion of blockchain in the energy sector from both technology and economy aspects. The Brooklyn microgrid project practices the application of blockchain for microgrid energy markets [21]. A blockchain-secured demand response scheme is proposed to promote individualized incentive pricing under a dual-incentive mechanism in [22]. Focusing on the resource-consuming drawbacks of blockchain itself, beneficial solutions to reduce frequent transactions on blockchain is proposed in [23].

However, the blockchain technology has its inherent weakness. In blockchain environment, smart contract can execute contract automatically and securely. For the sake of secure operation, some measures have been taken such as designing a so-called gasLimit variable in some blockchain environment, e.g., Ethereum. This variable restricts the number of computation steps, the logic of contract contents, and the complexity of contract logic. Hence, current smart contracts are able to support simple scripting language [24] but cannot support complex calculations. This limitation is very practical, considering that the consensus process also makes it difficult and unnecessary to include complex calculations. A consortium blockchain-enabled secure energy trading framework for electric vehicles is proposed in [25], and the contract optimization problem is solved by using the iterative convex–concave procedure algorithm. It demonstrates how to get contract items using off-chain computation, while not involving the implementation of smart contracts. A decentralized cooperative demand response framework is presented in [26] to manage the daily energy exchanges, and smart contract is utilized to enforce autonomous monitor and transaction. Currently, the respective on-chain and off-chain tasks and their cooperation in the blockchain environment are not widely discussed in the literature.

For remote wind farms with P2G systems which trade in multiple energy markets, an automated scheduling framework under blockchain-based smart contract is proposed in order to guarantee that transactions in multiple energy markets can still proceed under potential communication instability and insecurity at real-time schedule. Main contributions in this paper can be briefed as follows:

- A scheduling strategy considering the revenues of participating in multiple energy markets, the capability of reducing wind power curtailment, the penalizations of violating contract items, and the investment/operation cost of investing a wind farm equipping a P2G system is established, in which the non-linearity in the electrolysis of P2G system is considered with detailed models.
- An automated scheduling framework with both off-chain and on-chain procedures is proposed to ensure the applicability of smart contract in blockchain environment, especially in the case that the scheduling considers a non-linearity model of P2G system and trades in multiple energy markets.
- A modified smart contact protocol is adapted considering that more than one scheduling result from the wind farm can be submitted as potential contract items. Moreover, a two-stage scheduling processes and the off-chain/on-chain framework is simulated to compare the effectiveness of the proposed approach.

The rest of the paper is organized as follows. Section 2 describes the framework of smart-contract-enabled automatic scheduling for remote wind farms participating in multiple energy markets. Section 3 introduces the non-linear modeling and scheduling objectives for such a wind farm with the P2G system. Section 4 illustrates the implementation of the proposed framework with commonly used smart contracts. Section 5 provides results of simulations. Conclusions are discussed in Section 6.

## 2. Smart-Contract-Enabled Automated Scheduling Framework

For a wind farm, being equipped with a P2G can further help to enhance the scheduling economy, improve the wind power accommodation, and reduce carbon emissions. The amount of carbon P2G absorbs can be regarded as the permits of carbon emission P2G owns, which can be sold in carbon market. However, there are some concerns with this kind of scheduling involving multiple energy types. The first one is that energy markets are isolated from each other, which makes it difficult for wind farms to coordinate their scheduling plans. The second one is that, for remote wind farms, it is not easy to guarantee the stability and trustworthy of information considering the communication conditions.

Focusing on these concerns, a smart-contract-enabled automated scheduling framework for remote wind farms with P2G systems is established and the overall framework is shown in Figure 1. In blockchain environment, smart contract can only support simple scripting language considering the operation security. Due to the insufficient calculation capability of the smart contract, the scheduling objective with a non-linear P2G model are solved off-chain. The respective functionalities of on-chain and off-chain procedures are described as follows:

- The off-chain procedure is executed by the wind farm, and is able to find a set of potential scheduling results. Even without the framework proposed here, one wind farm is obliged to run a scheduling function and report its scheduling results in corresponding energy markets. In addition, since predictions on wind power output are often difficult to limit to one particular result, it is also very common to obtain a set of potential scheduling results based on multiple predicted wind power output curves. Although the objectives in [25] are electric vehicles, the process of obtaining results from off-chain procedure is similar to this paper. Details on obtaining contract items will be given in Section 3.
- The on-chain procedure is used to urge that one of these scheduling results can be recognized and executed between wind farms and multiple energy markets. Each participator in the blockchain—i.e., a wind farm owner and organizers of multiple energy markets—votes in one scheduling result from the set of potential scheduling

results, and automatically settles among participators based on the smart contract. Specifically, the Open Vote Network (OVN), i.e., a voting protocol as a smart contract in Ethereum [27], is adapted. Details on reclaiming this security and honesty through OVN will be explained in Section 4.

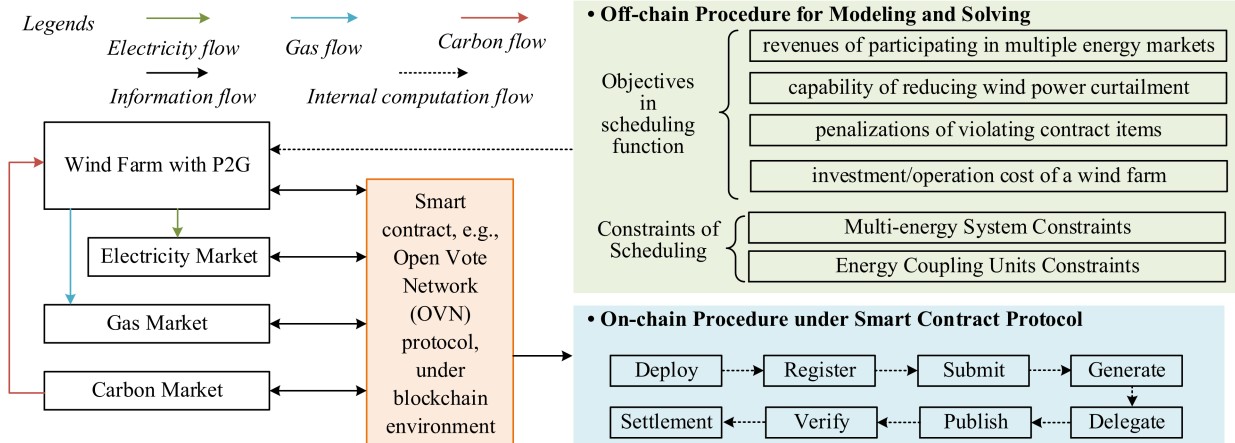

**Figure 1.** Framework of smart-contract-enabled automated scheduling for remote wind farms with P2G embedded considering multiple energy markets.

A typical two-stage scheduling process, i.e., day-ahead and real-time scheduling, is described for simplicity. In this framework, real-time scheduling can directly utilize the day-ahead scheduling result, eliminating the need for a new round of scheduling solving and confirmation with multiple energy markets. Moreover, in order to avoid that the retraction of deposits in the blockchain, the wind farm and multiple energy markets can trust each other to transfer a certain set of buying/selling volumes in real-time as agreed in the smart contract.

## 3. Off-Chain Modeling and Solving for Wind Farm with P2G System

In order to improve the accuracy of scheduling results, this section details the non-linear modeling of the P2G system. A scheduling objective function considering multiple energy markets is established for the whole wind farm with the P2G system.

### 3.1. Non-Linear Modeling of P2G System

Due to the uncertainty of wind power, bidding strategies, and congestion, a large amount of wind power is abandoned. In such a case, P2G system can sell an appropriate amount of carbon emission permits to obtain raw material of carbon, and can reduce wind power curtailment using electrolysis. The products of electrolysis can be combined with carbon to generate methane, which can be sold in the gas market. The P2G technology generally include two main steps of electrolysis ($2H_2O \rightarrow 2H_2 + O_2$) and methanation ($CO_2 + 4H_2 \rightarrow CH_4 + 2H_2O$). Typical energy conversion process of a P2G system is shown in Figure 2.

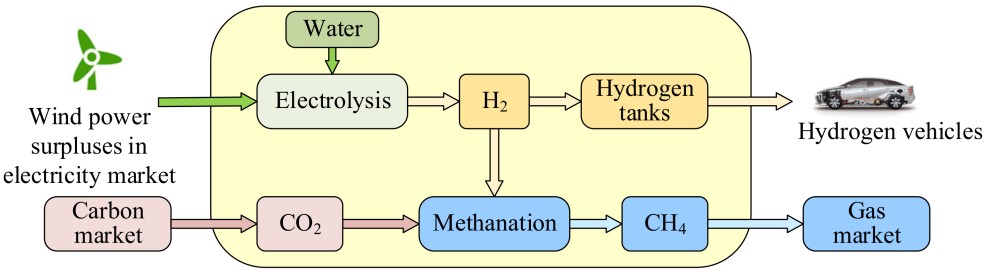

**Figure 2.** Diagram of energy conversion processes of a P2G system.

As shown in Figure 2, the first and crucial step is electrolysis, which consumes wind power to electrolyze water. The produced hydrogen can be pumped into hydrogen tanks for storage and to supply hydrogen loads—e.g., hydrogen powered vehicles. Meanwhile, an additional step of methanation is able to convert hydrogen and carbon into methane, which means this production process can absorb carbon. Therefore, the methanation process is considered in the P2G model in this paper, and all hydrogen produced by electrolysis is used to produce methane. All the rest units in the P2G system are summarized as the balance-of-plant (BoP) devices.

The proton exchange membrane electrolyze has the advantages of fast reaction, no pollution, and high efficiency, and thus is mostly used in recent engineering and research [28], which is also adopted for modeling in this paper. A detailed model considering the nonlinear nature of electrolysis is built as follows. It is composed of many electrolysis cells connected in series and in parallel. Through the continuous adjustment of cell currents, the electrolysis can promptly respond to the change of surplus wind powers, and accordingly adjust hydrogen production. For an electrolysis cell, its total voltage corresponds to the sum of open circuit voltage, activation overpotential, and ohmic overpotential, which is shown as

$$V_{\text{cell},t} = V_{\text{ocv},t} + \eta_{\text{act},t} + \eta_{\text{ohm},t} \tag{1}$$

where $V_{\text{cell},t}$ is the total voltage of an electrolysis cell; $V_{\text{OCV},t}$ is the open circuit voltage; $\eta_{\text{act},t}$ is the activation overpotential; $\eta_{\text{ohm},t}$ is the ohmic overpotential.

The open circuit voltage is calculated using Nernst equation [28,29], which is shown as

$$V_{\text{ocv},t} = V_{\text{eq}} + \frac{RT}{2F} \ln\left(\frac{p_{\text{H}_2} p_{\text{O}_2}^{1/2}}{p_{\text{H}_2\text{O}}}\right) \tag{2}$$

$$V_{\text{eq}} = 1.229 - 0.9 \times 10^{-3}(T - 298.15) \tag{3}$$

where $V_{\text{eq}}$ is an equilibrium voltage related to cell temperature; $R$ is the universal gas constant; $T$ is the cell temperature; $F$ is the Faraday constant; $p_{\text{H}_2}$, $p_{\text{O}_2}$ and $p_{\text{H}_2\text{O}}$ are partial pressures of $\text{H}_2$, $\text{H}_2\text{O}$, and $\text{O}_2$, respectively.

The activation overpotential involves overcoming energy barriers at the reaction site. The relationship between the activation overpotential and current density is commonly described using Butler–Volmer equation [29], which is shown as

$$\eta_{\text{act},t} = \frac{RT}{\alpha_{\text{a}}F} \text{arsinh}\left(\frac{i_{\text{cell},t}}{2i_{\text{a}}}\right) + \frac{RT}{\alpha_{\text{c}}F} \text{arsinh}\left(\frac{i_{\text{cell},t}}{2i_{\text{c}}}\right) \tag{4}$$

where $\alpha_{\text{a}}$ and $\alpha_{\text{c}}$ are charge transfer coefficients at the electrodes of anode and cathode; $i_{\text{cell},t}$ is the cell current; $i_{\text{a}}$ and $i_{\text{c}}$ are exchange current densities at anode and cathode.

The ohmic overpotential occurs due to the electrical resistance of the electrolysis cell, which can be calculated by Ohm's law as

$$\eta_{\text{ohm},t} = (R_{\text{pem}} + R_{\text{con}})i_{\text{cell},t} = \left(\frac{\rho_{\text{pem}}d}{A_{\text{cell}}} + R_{\text{con}}\right)i_{\text{cell},t} \tag{5}$$

where $R_{\text{pem}}$ and $R_{\text{con}}$ are resistances of the proton exchange membrane and connections; $\rho_{\text{pem}}$ is the resistivity of the proton exchange membrane; $d$ is the thickness of the membrane; $A_{\text{cell}}$ is the area of the membrane.

According to the connection mode of electrolysis cells and the relationship between the cell current and cell voltage, the power consumption and the amount of hydrogen production of the electrolysis are shown as

$$P_{\text{E},t} = \mu_1 N_{\text{stack}} N_{\text{cell}} A_{\text{cell}} V_{\text{cell},t} i_{\text{cell},t} \tag{6}$$

$$w_{\text{E},\text{H}_2,t} = \mu_2 N_{\text{stack}} N_{\text{cell}} A_{\text{cell}} \frac{\eta_f i_{\text{cell},t}}{2F} \tag{7}$$

where $\mu_1$ and $\mu_2$ are conversion factors; $N_{\text{stack}}$ and $N_{\text{cell}}$ are the number of parallel and series electrolysis cells, respectively; $\eta_f$ is the Faraday efficiency.

The relationship between the efficiency of hydrogen production and the amount of hydrogen production is shown as

$$\eta_{E,H_2,t} = \frac{L_{\text{HHV},H_2} w_{E,H_2,t} \Delta t}{\mu_3 P_{E,t} \Delta t} = \frac{L_{\text{HHV},H_2} \mu_2 \eta_f}{2\mu_1 \mu_3 F V_{\text{cell},t}} \tag{8}$$

where $L_{\text{HHV,H2}}$ is the higher heating value of hydrogen; $\mu_3$ is a conversion factor.

The amount of methane production and carbon consumption of a methanation system are depicted as

$$w_{M,CH_4,t} = \frac{\mu_4 \eta_{M,CH_4} w_{E,H_2,t}}{\rho_{CH_4}} \tag{9}$$

$$w_{M,CO_2,t} = \frac{w_{E,H_2,t} M_{CO_2}}{\mu_5 \rho_{CO_2}} \tag{10}$$

where $\eta_{M,CH_4}$ is the efficiency of methane production; $\rho_{CH_4}$ is the density of methane at standard atmospheric pressure; $M_{CO_2}$ is the molar mass of carbon; $\rho_{CO_2}$ is the density of carbon at standard atmospheric pressure; $\mu_4$ and $\mu_5$ are molar conversion factors.

### 3.2. Objective Function

For a wind farm with P2G system embedded, the economic parameters include investment costs and operational costs for wind turbine generators (WTGs) and P2G equipment (i.e., electrolysis, methanation, and BoP devices). The cost of the system in a scheduling day is calculated as [30,31].

$$SIO = \sum_{i=1}^{I} \frac{CAP_i INVE_i \frac{\tau(1+\tau)^{y_i}}{(1+\tau)^{y_i}-1} + OPEX_i}{DAY} \tag{11}$$

where $i$ is a component index (e.g., WTGs, electrolysis, methanation, and the BoP devices); $CAP_i$ is the capacity of component $i$; $INVE_i$ is the investment cost of component $i$; $\tau$ is the interest rate; $y_i$ is the lifetime of component $i$; $OPEX_i$ is the operation and maintenance expenditure of component $i$ for a year; $DAY$ is the number of available days in a year.

The P2G system has the ability to reduce emissions of carbon. From the carbon market perspective, the amount of carbon the P2G system absorbs can be regarded as the permits of carbon emission owned by the P2G system, which can be sold in carbon market to gain revenue. Therefore, the process of P2G participating in the carbon market can be seen as a process of selling carbon emission permits and gaining revenue.

The typical wind power curves can be generated by clustering. A simple example to select a typical wind power curve $k_0$ from some possible wind power outputs is according to the following judgement.

$$\sum_{t=1}^{T} \left(P_{\text{tyw},k_0,t} - P_{\text{pw},t}\right)^2 \le \sum_{t=1}^{T} \left(P_{\text{tyw},k,t} - P_{\text{pw},t}\right)^2, \forall k \in K \tag{12}$$

where $P_{\text{tyw},k,t}$ is the $k$th possible wind power output at period interval $t$; $P_{\text{pw},t}$ is the predicted wind power at period interval $t$; $K$ is the set of typical wind power curves.

Then, the corresponding scheduling plans are executed as the optimal scheduling plans of the predicted wind power curves. Considering the day-ahead prediction deviations, there are penalizations for violating the contract items of executing scheduling plans of wind power curve $k_0$. The default volume of energy in corresponding market is shown as

$$p_{j,\text{def},t} = \left| p_{j,t} - p_{j,k_0,t} \right| \tag{13}$$

where $j$ is an energy market index (e.g., electricity, gas, and carbon market); $p_{j,t}$ is the volume of energy trading in energy market $j$ at period interval $t$; $p_{j,k_0,t}$ is the volume of energy sold in energy market $j$ according to the scheduling plans of wind power curve $k_0$.

A comprehensive scheduling objective for the remote wind farm considers the revenues of participating in multiple energy markets, the capability of reducing wind power curtailment, the penalizations of violating contract items, and the investment/operation cost of a wind farm equipped with a P2G system. The revenues of participating in multiple energy markets is characterized from energy trading. The capability of reducing wind power curtailment is characterized by the cost for wind power curtailment. The penalized deposit of violating contract items is characterized by the smart contract agreed between the wind farm and energy markets. Optimizing the scheduling objective aims to maximize the total net revenue in a scheduling day, which can be expressed as

$$\max Y = \sum_{t=1}^{T} \sum_{j=1}^{J} \left( c_j p_{j,t} - c_{j,\text{def}} p_{j,\text{def},t} \right) - \sum_{t=1}^{T} c_{\text{wp,curt}} p_{\text{wp,curt},t} - SIO \tag{14}$$

where $c_j$ is the energy price in energy market $j$; $c_{j,\text{def}}$ is the penalized deposit of violating contract items; $c_{\text{wp,curt}}$ is the cost for wind power curtailment; $p_{\text{wp,curt},t}$ is the volume of wind power curtailment at period interval $t$.

### 3.3. Constraints

The constraints of P2G are depicted as

$$0 \leq P_{\text{E},t} \leq P_{\text{E,rated}} \tag{15}$$

$$-P_{\text{E,down}} \leq P_{\text{E},t-1} - P_{\text{E},t-1} \leq P_{\text{E,up}} \tag{16}$$

$$i_{\text{cell,min}} \leq i_{\text{cell},t} \leq i_{\text{cell,max}} \tag{17}$$

where $P_{\text{E,rated}}$ is the rated input power of the electrolysis; $P_{\text{E,down}}$ and $P_{\text{E,up}}$ are downward ramping rate and upward ramping rate, respectively; $i_{\text{cell,min}}$ and $i_{\text{cell,max}}$ are the minimum and maximum cell current.

The constraint of electricity network is shown as

$$P_{\text{e,min}} \leq p_{\text{electricity},t} \leq P_{\text{e,max}} \tag{18}$$

where $P_{\text{e,min}}$ and $P_{\text{e,max}}$ are the minimum and maximum grid-connected power, respectively.

The congestion constraints of gas network are depicted as

$$0 \leq w_{\text{M,CH4},t} \leq Q_{\text{g,max}} \tag{19}$$

$$0 \leq \sum_{t=1}^{T} w_{\text{M,CH}_4,t} \leq C_{\text{g,max}} \tag{20}$$

where $Q_{\text{g,max}}$ is the maximum gas flow, $C_{\text{g,max}}$ is the upper limit of total flow on a trading day.

It should be noted that there are some simplifications in the objective function and the constraints:

1.  Since methane can be pumped directly into existing natural gas pipelines for large-scale storage and long-distance transmission, the economic costs associated with constructing pipelines are not considered in Equation (11).
2.  The wind farm participates in multiple energy markets as price takers, i.e., values of $c_j$ in Equation (14) are parameters other than variables.
3.  The wind farm is connected to the electricity/gas market by a single line/gas pipeline, as implied in Equations (18)–(20).

## 4. Implementation of Smart Contract under Blockchain Environment

### 4.1. Structure of Implemention

In general, a smart contract is a piece of codes running on blockchain environment whose logic defines it content. It can be used to receive and process information, store and transfer assets, etc. [32]. It is immutable once deployed and will remain dormant until some transactions submitted by a participator's account triggers it. A smart contract can contain a wide variety of contract items [25]. Therefore, for the implementation here, typical wind power curves and corresponding scheduling results in multiple energy markets can be defined as contract items.

Smart contracts are different from traditional programming language in that they are essentially a kind of agreement for the transfer of digital assets between untrusted participants. The simpler the code, the more secure and reliable it is. The smart contract is thus designed to support only simple scripting language in a particular environment—e.g., blockchain—and cannot support complex calculations. Considering the insufficient calculation capability of smart contracts, the complex modeling and solving process in Section 2 is implemented as an off-chain process, while only the process including submitting and confirming of energy transactions are put on-chain, as shown in Figure 3.

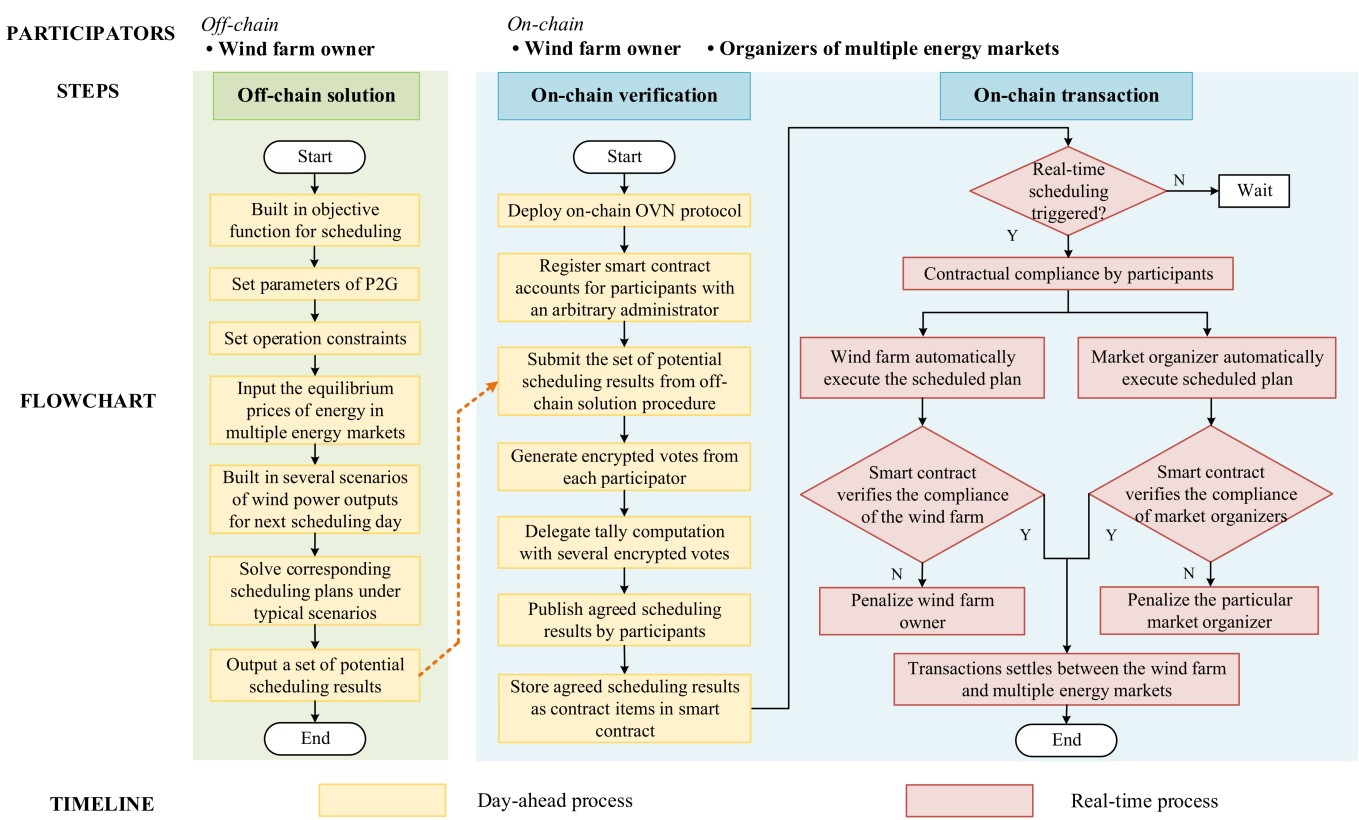

**Figure 3.** Processes in implementing the smart-contract-enabled automatic scheduling for a wind farm with P2G system in multiple energy markets.

Unlike the traditional scheduling approach, if a blockchain-backed smart contract is introduced, it will bring at least two benefits. On one hand, the traditional centralized multi-level structure among multiple markets to coordinate scheduling plans is changed. Under the blockchain environment, remote wind farms with P2G systems, as well as market organizers—i.e., electricity, gas, and carbon markets—can work as decentralized nodes. Wind farm owners, along with organizers of multiple energy markets, launch votes on a set of potential scheduling plans. With this structure, no one participant is the default dominance, guaranteeing fairness and reducing the need for complex collaboration. On the other hand, these transactions are arbitrated and recorded on-chain, and no matter what

communication problems occur in real-time, the relevant transactions can be automatically executed in the same way as voted upon a day ahead, which can guarantee the timeliness and accuracy of real-time scheduling.

### 4.2. Off-Chain Procedure for Modeling and Solving

In the day-ahead process timeline, the wind farm collects a set of predicted wind power curves as well as equilibrium energy prices from multiple energy markets and operating parameters for power transmission lines and gas pipelines. As shown in Figure 3, the optimized scheduling objectives are modeled and solved off-chain by the wind farm alone.

### 4.3. On-Chain Procedure under Smart Contract Protocol

The on-chain processes in the automated scheduling for remote wind farms and multiple energy markets are divided into two timelines, which are given as follows.

#### 4.3.1. Day-Ahead On-Chain Processes

The day-ahead on-chain processes focus on setting up the smart contract.

- Deployment of a smart contract protocol—Several smart contract protocols have been developed for different applications. The OVN protocol is able to provide a public bulletin board in a decentralized internet to support coordination among multiple participators [33]. All computations in OVN are written as a smart contract. The following processes are mainly developed under a standard OVN but with necessary modifications.
- Registration and deposition of participators—Like a permissioned blockchain, OVN-based smart contract only allows eligible participators. Although an administrator is required by the OVN protocol to authorize accounts, it is not necessarily a trusted authority. The following provision sets an arbitrary organizer from multiple energy markets as this administrator. The wind farm owner and other market organizers register as accounts participating in the smart contract.
- Submission of potential scheduling results as voting keys—The wind farm owner submits a set of scheduling results based on its off-chain solving. Through the restriction of a smart contract, dishonesty about how much energy the wind farm can provide will only result in penalties for the wind farm itself not being able to provide/absorb the corresponding physical energy, and the consideration of this kind of penalty is included in the objective function Equation (14).
- Generation of potential scheduling results as votes—After voting keys are submitted by the wind farm, all participators—i.e., market operators and wind farm itself— generate and broadcast their respective votes to the other nodes. If needed, an encryption can make the selections of participators anonymous and immutable along during broadcast among participators.
- Delegation and storage of selected scheduling result—The administrator delegates and publish all participators' votes, and all participators can examine as they wanted. The final voted scheduling result is casted as the contract items that stored in the smart contract.

Then, the on-chain process hangs until real-time scheduling triggers the next step.

#### 4.3.2. Real-Time On-Chain Processes

The real-time on-chain processes focus on executing smart contract under the voted scheduling plan:

- Automated real-time schedule by participants—In the real-time schedule, any participant can automatically schedule based on contract items that have been agreed on-chain in day-ahead processes. However, defaults may happen. A typical default situation for a wind farm is that it fails to buy/sell the agreed energy volumes in

a corresponding market. A typical default situation for a market is the inability to receive/supply the agreed energy volumes because of line constraints.
- Verification of compliance on energy trades by smart contract—The smart contract verifies whether the participants have strictly executed contract items. Unlike purely digital assets, energy volumes can be physical measured and difficult to tamper with. Moreover, in such a framework, even if information instability of remote wind farms occurs—e.g., delays—it only affects the settlement time of smart contract and not the timeliness of real-time scheduling.
- Settlement among participators—When all the scheduling hours of the real-time schedule finish, electricity, gas, and carbon markets settles with the wind farm respectively, including penalties for violating the agreed contract items.

## 5. Case Study

### 5.1. Parameter Settings

In this section, a 40MW wind farm with a 6MW P2G system is taken as an example to verify the effectiveness of the proposed scheduling approach. The day-ahead wind power output curve in the wind farm for real-time scheduling is shown in Figure A1 in Appendix A, which is adapted from [34] with some scaling according to their rated powers. Several typical day-ahead predicted curves of wind power output as the potential contract items are shown in Figure A2 in Appendix A, which are from [35] and some modifications have been made on this basis. Day-ahead scheduling results of the remote wind farm with a P2G system in multiple energy markets including electricity, gas, and carbon are solved off-chain by the wind farm owner. Four cases are presented to illustrate the effectiveness of the proposed scheduling model as follows.

Case 1: No P2G system is utilized for the wind farm, and thus trades only exists in the electricity market.

Case 2: The P2G system is utilized for the wind farm, and is described with simplified model as given in [36]. The adoption of smart contracts is not considered.

Case 3: The P2G system is utilized for the wind farm, the non-linearity nature of electrolysis is considered. The adoption of smart contracts is not considered.

Case 4: The P2G system is utilized for the wind farm, while the non-linear nature of electrolysis is considered. the adoption of smart contracts among all the market operators and the wind farm owner is considered.

The parameters of P2G system obtained from [37] are shown in Table 1. Economic parameters for WTGs, electrolysis, methanation, and BoP devices are shown in Table 2, which are obtained from [30,31] and assumptions. The equilibrium price in the electricity market and its penalty in the smart contract are set as 0.4 ¥/kWh and 2.6 ¥/kWh, respectively. The equilibrium price of methane and its penalty are set as 2.56 ¥/Nm$^3$ and 16.9 ¥/Nm$^3$, respectively. The equilibrium price of carbon and its penalty are set as 0.59 ¥/Nm$^3$ and 19.5 ¥/Nm$^3$, respectively. The cost for curtailing wind power is set as 1.2 ¥/kWh. The scheduling period is 24-h with a 1-h time interval.

**Table 1.** Parameters of P2G system.

| Parameters | | | |
|---|---|---|---|
| | $T = 335.15$ K | $R = 8.314$ J/mol·K | $F = 96{,}485$ C/mol | $p_{H2} = 29.8$ bar |
| | $p_{O2} = 2.8$ bar | $p_{H2O} = 1$ bar | $\alpha_a = 2$ | $\alpha_c = 0.5$ |
| | $i_a = 1 \times 10^{-6}$ A/cm$^2$ | $i_c = 1 \times 10^{-3}$ A/cm$^2$ | $R_{pem} + R_{con} = 0.12$ R·cm$^2$ | $\mu_1 = 0.001$ |
| P2G | $\mu_2 = 3.6$ | $\mu_3 = 3600$ | $\mu_4 = \mu_5 = 4$ | $N_{stack} = 3$ |
| | $N_{cell} = 250$ | $A_{cell} = 1100$ cm$^2$ | $\rho_{CH4} = 0.7174$ kg/m$^3$ | $\eta_f = 99\%$ |
| | $\rho_{CO2} = 1.977$ kg/m$^3$ | $i_{cell, min} = 0.15$ A/cm$^2$ | $i_{cell, max} = 3$ A/cm$^2$ | |

**Table 2.** Economic parameters for WTGs, electrolysis, methanation, and BoP devices in the wind farm.

| Component $i$ | $CAP_i$ (MW) | $INVE_i$ (¥/kW) | $\tau$ (%) | $y_i$ (year) | $OPEX_i$ (% of $CAP_iINVE_i$) |
|---|---|---|---|---|---|
| WTGs | 40 | 3500 | 7 | 20 | 2.75 |
| Electrolysis | 6 | 4000 | 7 | 20 | 2.75 |
| Methanation | 4.5 | 3500 | 7 | 20 | 2.75 |
| BoP devices | 3 | 3000 | 7 | 20 | 2.75 |

### 5.2. Analysis of Scheduling Results among Different Cases

Before activating the smart contract protocol (i.e., OVN), potential contract items (i.e., multiple energy volumes that will be trade and record in real-time energy markets) are calculated off-chain by the wind farm owner are obtained, which are shown in Figure 4. Considering the infeasibility of complex calculations in smart contract, the on-chain OVN only needs to claim some typical wind power curves $k_0$ based on (12) and its corresponding scheduling plans among multiple market operators and the wind farm owner. In this paper, the claimed typical wind power curve $k_0$, which satisfies Equation (12), is $k_0 = 3$.

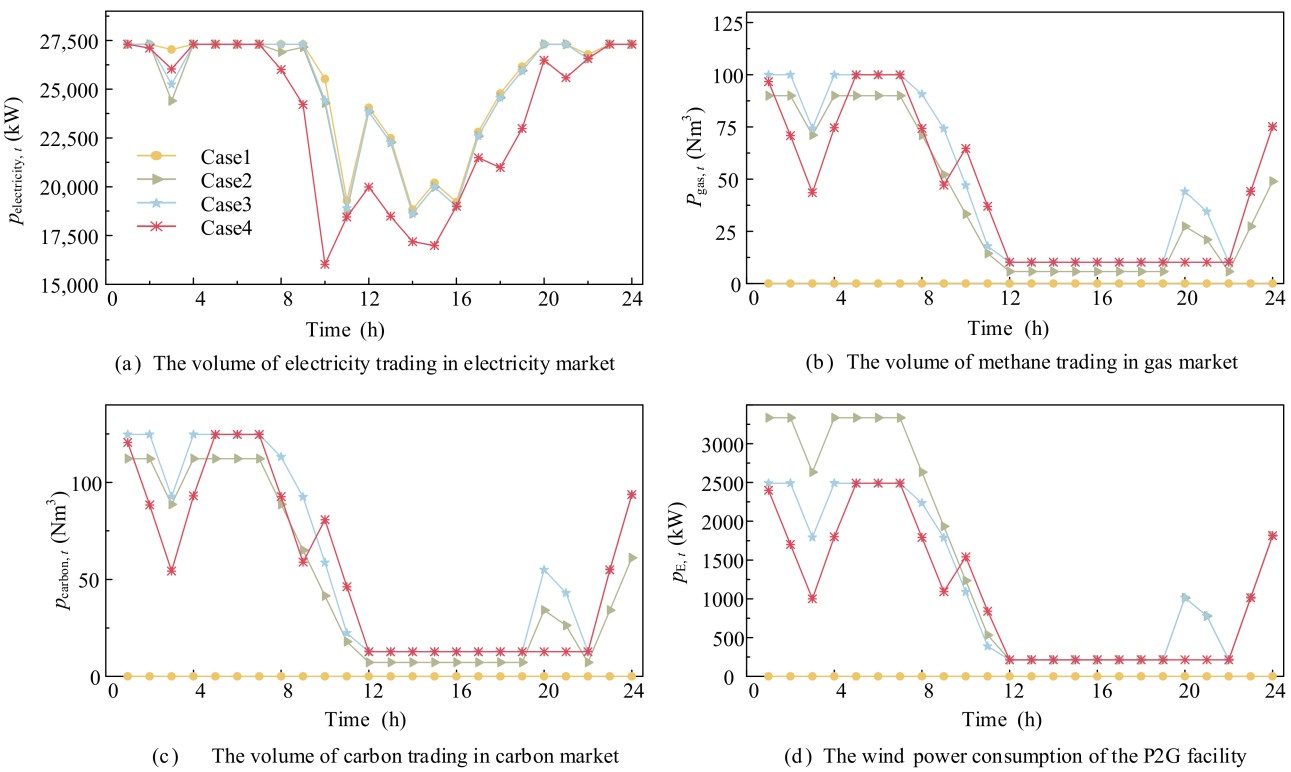

(a) The volume of electricity trading in electricity market

(b) The volume of methane trading in gas market

(c) The volume of carbon trading in carbon market

(d) The wind power consumption of the P2G facility

**Figure 4.** Day-ahead optimal scheduling results of P2G in remote wind farms considering multiple energy markets.

In multiple energy markets, the day-ahead scheduling results of the wind farm with a P2G system embedded are calculated off the chain in above four cases, respectively. The optimal scheduling results of P2G in remote wind farms in each time period are shown in Figure 4. There are some differences between the scheduling results of different cases. Based on the deviation, this section has detailed analyses on the capability of P2G system, the non-linearity nature of electrolysis and the performance of smart contract.

#### 5.2.1. Analysis on the Capability of P2G System

The revenue of the remote wind farm equipped with a P2G system from multiple energy markets and wind power curtailment rate are shown in Table 3. In Case 1, due to the lack of a P2G system, the wind farm only trades with electricity market, and the surplus wind power cannot be absorbed to generate methane and reduce carbon emissions. In Cases 2, 3, and 4, due to the embedding of the P2G system, surplus wind power has a way to be absorbed. The steps of electrolysis and methanation are carried out in turn to generate

methane and consume carbon by absorbing surplus wind power. Thus, the wind farm with a P2G system embedded can have energy trading with electricity, gas, and carbon markets, respectively, which has obvious economic benefits and carbon consumption effect.

**Table 3.** Revenues from multiple energy markets and effect on wind power curtailment reduction rate.

| Case | Revenue from Electricity Market (¥) | Revenue from Gas Market (¥) | Revenue from Carbon Market (¥) | Wind Power Curtailment Reduction Rate (%) |
|---|---|---|---|---|
| 1 | 241,900 | / | / | 9.56 |
| 2 | 239,152 | 2455 | 706 | 5.28 |
| 3 | 239,824 | 3056 | 879 | 6.03 |
| 4 | 225,865 | 2664 | 766 | 11.84 |

### 5.2.2. Analysis on the Non-Linearity Nature of Electrolysis

Operation situations of a P2G system which consist of power input, hydrogen production, methane production, and carbon consumption are shown in Table 4. In Case 2, the P2G system obviously assumes more surplus wind power than Case 3, while its hydrogen production, methane production, and carbon consumption are less than Case 3. The reason is that Case 2 crudely considers that the relationship between hydrogen production and power consumption is simple and linear, and it ignores the non-linearity nature of electrolysis. The hydrogen production efficiency and the hydrogen output deviation between the model used in this paper and the crude linear model are shown in Figure 5. Although the hydrogen production efficiency of the two models is equal when the power input of the electrolysis is at a certain value, the overall hydrogen production efficiency of the two models is still obviously different. With the increase of power input, the hydrogen production deviation shows a trend of first increasing, then decreasing, and finally increasing, which exactly reflects the complexity of the electrolysis model. The crude model is insufficient to reflect the complex and non-linear nature of P2G facilities, consequently leading to inaccurate scheduling results.

**Table 4.** Operation situations of a P2G system.

| Case | $P_E$ (MW) | $w_{E,H2}$ (Nm³) | $w_{M,CH4}$ (Nm³) | $w_{M,CO2}$ (Nm³) |
|---|---|---|---|---|
| 1 | / | / | / | / |
| 2 | 35.52 | 4776.96 | 958.85 | 1196.05 |
| 3 | 28.78 | 5947.19 | 1193.75 | 1489.05 |
| 4 | 24.82 | 5183.44 | 1040.45 | 1297.83 |

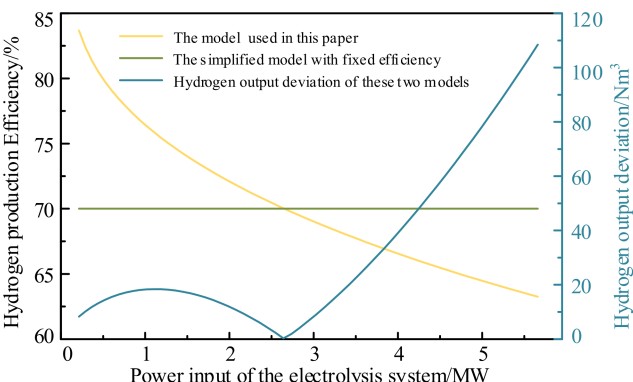

**Figure 5.** Hydrogen production efficiency and the hydrogen output deviation between the non-linear model in this paper and the simplified model.

### 5.2.3. Analysis of the Performance of Adopting Smart Contract

The typical wind power curve $k_0$ and the scheduling results under voted scheduling plan in Case 4 are shown in Figure 6. Caused by the differences between the typical wind

power curve as the contract items and the predicted wind power curve at each hour, the scheduling results can be slightly different. The performance of adopting smart contract is shown in Table 5. When any participator violates the voted scheduling plan, a penalty is costed as the characteristics of a smart contract. With this type of smart-contract-enabled penalty mechanism in Case 4, the total penalized cost $C_{AC}$ is much lower than Cases 1, 2, and 3, while the wind power curtailment rate is higher than Cases 2 and 3. According to the economic parameters in Table 2, the cost of the wind farm in the scheduling day is SIO = ¥76,691. The total net revenue $Y$ thus can be obtained as shown in the last column in Table 5.

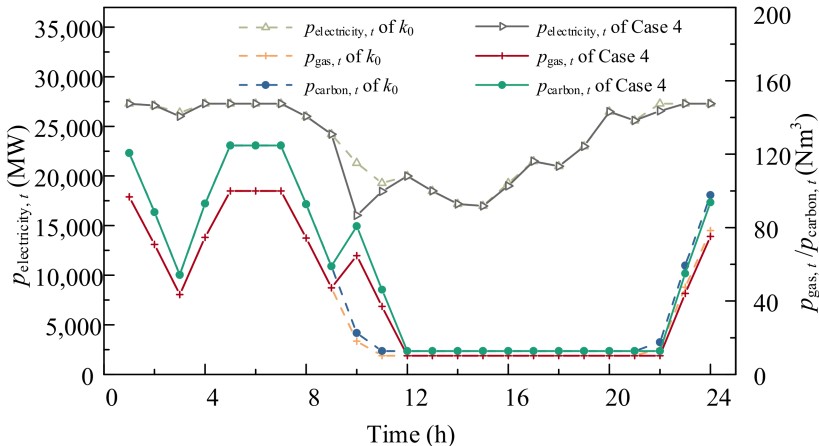

**Figure 6.** Scheduling plans of typical wind power curve $k_0$ and the scheduling results of Case 4.

**Table 5.** Performance of adopting smart contract in different cases.

| Case | Penalization of Wind Farm (¥) | Violation of Electricity (MW) | Violation of Methane (Nm$^3$) | Violation of Carbon (Nm$^3$) | $C_{AC}$ (¥) | $Y$ (¥) |
|---|---|---|---|---|---|---|
| 1 | 76,710 | 32.58 | 977 | 1219.27 | 125,000 | −36,501 |
| 2 | 42,337 | 35.03 | 218.43 | 272.47 | 100,073 | 23,210 |
| 3 | 48,401 | 32.52 | 237.33 | 296.04 | 94,342 | 24,324 |
| 4 | 95,035 | 7.51 | 84.03 | 104.81 | 22,993 | 34,576 |

To illustrate the impact of penalizations for violating the voted scheduling plan in the smart contract, a comparison between the values of penalty $C_{AC}$ and $Y$ between Case 3 and Case 4 are shown in Table 6. The first column shows how much the penalties scale compared with the three original values defined in Section 5.1. From Table 5, it is observed that a penalty mechanism impacts the performance of adopting smart contract significantly. From Table 6, it is further observed that when this penalty for violating contract items is small, the effect of considering the performance of adopting smart contract during scheduling on $Y$ is not significant, or even harms its effectiveness. In practical uses, the values of penalizations in the smart contract can be selected by experience and more tests.

**Table 6.** Performance under different penalties of violating contract items.

| Scale with the Reference Value of Penalities $c_j$ | Cost Item | Case 3 | Case 4 |
|---|---|---|---|
| ×0.8 | $C_{AC}$ (¥) | 75,474 | 55,936 |
|      | $Y$ (¥) | 43,192 | 41,801 |
| ×1 | $C_{AC}$ (¥) | 94,342 | 22,993 |
|    | $Y$ (¥) | 24,324 | 34,576 |
| ×1.2 | $C_{AC}$ (¥) | 113,211 | 83,097 |
|      | $Y$ (¥) | 54,550 | 56,673 |
| ×1.5 | $C_{AC}$ (¥) | 141,514 | 69,029 |
|      | $Y$ (¥) | −22,848 | 57,745 |
| ×2 | $C_{AC}$ (¥) | 188,685 | 92,038 |
|    | $Y$ (¥) | −70,019 | 55,444 |

### 5.2.4. Analysis on Investment and Return

Taking Case 4 as an example, the total net revenue in an available day as well as the sum of investment cost and operational cost for wind farm, electrolysis, methanation, and the BoP devices are shown in Figure 7. From the perspective of investment return, the income of the wind farm with a P2G system embedded is sufficient to pay the cost of the system in a day, which is economically feasible. It can be calculated that it takes 14.95 years to recover the investment cost of WTGs and the P2G system. Although the payback years are not few, different researchers have shown that the situation can change in 10–20 years [38,39]. Considering the advantages of P2G in enhancing the economic and decarbonizing benefits, wind farms with P2G is worthy to invest.

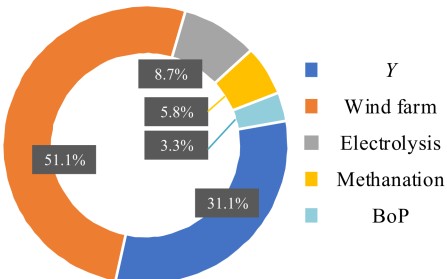

**Figure 7.** Total net revenue as well as the sum of investment cost and operational cost for WTGs, electrolysis, methanation, and BoP devices.

### 6. Conclusions

Predictions of uncontrollable wind power outputs are often not accurate enough, and the curtailment affects the economics of wind farms. By deploying P2G, wind farms can not only benefit from participating in multiple energy markets, but also contribute more for carbon reduction. An automated scheduling approach for remote wind farms equipped with P2G systems considering multiple energy markets is proposed in this paper in the presence of instable and unreliable information. Moreover, considering the insufficient calculation capability of smart contracts, a structure of off-chain solving and on-chain transaction is further developed. According to the simulation results, the main conclusions are summarized as follows:

1.  The results verify the effectiveness of the non-linear model of the P2G system. The electrolysis process is full of complexity and non-linearity, which should be taken into account when constructing the P2G model to improve accuracy of scheduling results.
2.  The proposed framework can cope with the limited complexity of smart contracts and insufficient computation. Specifically, off-chain solving is able to use a non-linear P2G model to obtain more accurate results, while the on-chain protocol only needs to consider a small set of potential scheduling plans.

3.  The proposed approach can effectively make full use of remote wind farms with P2G equipped—i.e., improve the economics of scheduling while reducing wind curtailment and decarbonization—while the execution of real-time scheduling can be ensured by smart contract items agreed a day ahead.

This paper is an exploration of adapting the fast-developing blockchain technology in the field of energy trading in multiple energy markets. For future research, the market behaviors from the multiple energy markets will be considered. Further verification will be done on blockchain-based platforms to capitalize energy trading. In addition, more market-realistic situations, such as more than one remote wind farms equipped with P2G systems, will be studied.

**Author Contributions:** Conceptualization, Z.G. and Z.J.; Data curation, Z.G. and Z.J.; Formal analysis, Z.J. and H.L.; Funding acquisition, Z.J.; Methodology, Z.G. and Z.J.; Validation, Z.G. and H.L.; Investigation, Z.G., and Z.J.; Writing—original draft, Z.G. and Z.J.; Writing—review and editing, Q.W. and H.L.; Supervision, Q.W.; Visualization, Z.G. and H.L. All authors have read and agreed to the published version of the manuscript.

**Funding:** This research was funded by National Natural Science Foundation of China (52107100), Natural Science Foundation of Jiangsu Province (BK20190710), General Project of Natural Science Research in Colleges and Universities of Jiangsu Province (19KJD470004) and Key Research and Development Program of Jiangsu Province (BE2020081-4).

**Institutional Review Board Statement:** Not applicable.

**Informed Consent Statement:** Not applicable.

**Data Availability Statement:** Not applicable.

**Conflicts of Interest:** The authors declare no conflict of interest.

**Appendix A**

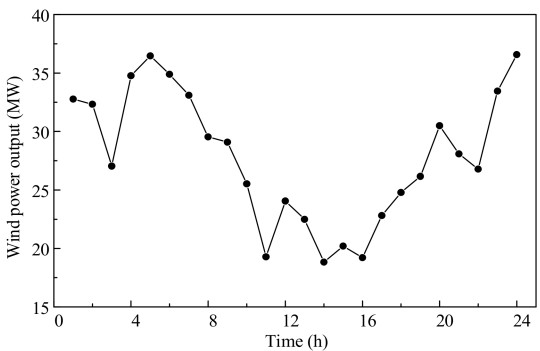

**Figure A1.** Day-ahead wind power output curve.

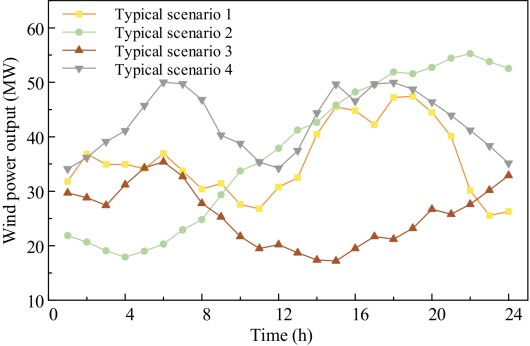

**Figure A2.** Several typical day-ahead predicted wind power curves.

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
