# Peer review of "Automated Scheduling Approach under Smart Contract for Remote Wind Farms with Power-to-Gas Systems in Multiple Energy Markets"

_energies, doi:10.3390/en14206781_

Round 1

Reviewer 1 Report

Dear authors,

The research paper called Automated Scheduling under Blockchain Environment for Power-to-gas Systems in Remote Wind Farms Considering Multiple Energy Markets is really very interesting for the readers, well done and  original. 

The significance of the content might be interesting for producers, which is a high merit for you.

I appreciated also the fact that the scientific papers used for references are quite new and published in significant journals, very well indexed.

I will give your paper an high overall merit.

Please,  make more papers like this one. I liked it a lot ! And continue with your research in the field.

Author Response

Dear reviewer,

Thank you very much for your valuable time and the highly comments you have given to this manuscript. It is the authors’ honor, and the authors hope to continue improving their abilities under your encouragement.

Reviewer 2 Report

The basic idea of the blockchain aspect is summarized in lines 294 to 297 -  that blockchain smart contracts are simple and the complex computation needs to be done off-chain.

This concept is already well-known in blockchain.

If a practical implementation of this concept is to be illuminated through the paper, then flow has to be indicated to be the initiation of the smart contract, then the feed of the input parameters every 24 h (line 365) and the storage of the computation result. It cannot be all mashed together as a single run-time activity as Figure 3 seems to suggest. For the initiation of the smart contract, the relevant aspects such as how many traders are involved in that single wind farm, who are the traders, how do they agree to the smart contract has to be indicated. For the computation every 24 h, how do the trading partners  provide the inputs, whether there is any validation and verification of inputs takes place, whether some inputs can be missing, and the impact of delays has to be indicated. For the storage of the computation, how the trading partners access the results of the computation should be brought out. If the wind farm is remote (line 29) where are the blockchain nodes - all of them? If the implementation is on Ethereum then that needs to be brought out.

On the other hand, If the authors only want to focus on the model then there is not much point in the blockchain aspect.

Author Response

Dear reviewer,

Thank you very much for your important and valuable comments on the manuscript. The authors value these comments and have carefully revised related sections on the basis. Descriptions of the relevant responses are as follows.

Comment 1

The basic idea of the blockchain aspect is summarized in lines 294 to 297 -  that blockchain smart contracts are simple and the complex computation needs to be done off-chain.

This concept is already well-known in blockchain.

If a practical implementation of this concept is to be illuminated through the paper, then flow has to be indicated to be the initiation of the smart contract, then the feed of the input parameters every 24 h (line 365) and the storage of the computation result. It cannot be all mashed together as a single run-time activity as Figure 3 seems to suggest. For the initiation of the smart contract, the relevant aspects such as how many traders are involved in that single wind farm, who are the traders, how do they agree to the smart contract has to be indicated. For the computation every 24 h, how do the trading partners  provide the inputs, whether there is any validation and verification of inputs takes place, whether some inputs can be missing, and the impact of delays has to be indicated. For the storage of the computation, how the trading partners access the results of the computation should be brought out. If the wind farm is remote (line 29) where are the blockchain nodes - all of them? If the implementation is on Ethereum then that needs to be brought out.

Response 1

Thanks a lot for this comment and suggestions on this essential technical point. Based on this valuable comment, authors have reorganized Section 4 to a large extent, as well as other relevant parts. Several important modifications are introduced as below.

(1) In the reorganized Section 4, Figure 3 is clearly modified, especially to make significant distinctions between different tasks about on-chain and off-chain, and different activities about day-ahead and real-time schedule. The lack of the above-mentioned clear distinction in the original Figure 3, as pointed out by the reviewer, did easily cause a suggested single run-time activity. Many thanks to the reviewer for pointing this out and make suggestions.

(2) The implementation is indeed based on Ethereum, as the reviewers found out. In the first draft, the authors were not able to give the relevant details clearly, resulting in the need for the readers to understand themselves, which has also been improved in the revised draft. And thanks again for pointing this out. Further, the authors also clarify the smart contract protocol used in the Ethereum environment, i.e., the Open Vote Network protocol, and give steps on how to implement the protocol to this automated scheduling framework. This protocol is able to deal with validations and verifications. Details can be found from lines 324 to 374.

(3) The wind farm and the multiple energy markets are nodes in the blockchain environment and are traders of the smart contract. Because day-ahead scheduling requires less immediacy, when a smart contract is started, it only needs to submit the result by a certain time deadline, which is the same as in the Open Vote Network protocol. Under the Open Vote Network protocol, the smart contract is agreed to by voting. When the real-time scheduling starts, the rest of the smart contract is triggered. And when there is any delays or missing inputs, etc., the scheduling of energy is carried out as agreed in the smart contract day-ahead, and the related settlement may wait for the completion of the information. This is mainly because the smart contract in this manuscript serves to automate the scheduling of remote wind farms under potentially inadequate information conditions and to urge both the wind farm and multiple energy markets to execute the relevant plans as agreed beforehand.

Thanks again for this valuable comment.

Comment 2

On the other hand, If the authors only want to focus on the model then there is not much point in the blockchain aspect.

Response 2

Many thanks to the reviewer for pointing this out. Indeed, one of the initial focuses in writing this manuscript is to develop a scheduling function in multiple energy markets for wind farms that include the nonlinear P2G model. With this valuable suggestion, the authors have better sorted out the logic and modified the relevant contributions (lines 104 to 116). Briefly, it starts with a scheduling function containing nonlinear models. Then, in an automated scheduling framework that includes on-chain and off-chain processes, the off-chain part is responsible for solving the model and the on-chain part is responsible for developing smart contracts between multiple markets and the wind farm. And considering that there are always more than one potential possibilities of wind power output predictions in day-ahead scheduling, the smart contract specifically considers a voting-included protocol.

Many thanks to the reviewer for the various valuable suggestions. The authors hope that the revisions can achieve some improvements.

Reviewer 3 Report

The authors propose a blockchain-based P2G scheduling for wind farms considering multiple energy markets. The overall discussion and presentation are complete from start to finish. However, some improvements are still required to enhance the quality of the paper further.

Regarding the use of blockchain in the paper

First of all, the authors propose an off-chain solution as a workaround for complex on-chain operations. However, how do the authors guarantee the security and fairness of this off-chain solution?

In lines 55-56, the authors said that "wind farms face the issue of insecure energy trading when participating in multiple energy markets. In addition, the electricity, gas, and carbon market are managed by different departments..." Considering this background, I cannot find solutions from the paper about the following issues.

  • Who should run the objective functions (Equation 11-14) as the off-chain solutions?
  • How can we trust that the person running those functions will always be honest and produce fair results?

Running the objective functions on-chain (as proposed by related works) can be considered secure and trustable because the smart contract execution is deterministic and hard-to-tamper due to the underlying consensus algorithm. However, if those functions are run off-chain, then how can you solve those questions?

Answering this question is crucial in a blockchain-based trading environment because participants are always untrusted and do not trust one another. Without answering this question, it is tough to pinpoint the usefulness of the blockchain in the paper. Mainly, how is the proposed system different if not using blockchain?

The contract items and contract status from Figure 1 are not explained. In addition, from my understanding, the contract items from Ref [25] are different meanings from the smart contract definition in general (e.g., the one in Ethereum). They are not the same.

Line 330-331, "It is worth noting that the performance of smart contract refers to the extent to which the scheduling plans of wind power curve k0 are fulfilled." This sentence is unclear. I think the authors need to describe better the "performance of smart contract" because it is used in their evaluations.

Usually, the "performance of a smart contract" is measured through a gas-used parameter, which as the authors mentioned, the simpler the better because of GasLimit. However, the authors define their own version of "performance of smart contract", which I cannot understand in this current manuscript due to lack of description.

Regarding implementation and evaluation

Where do the authors get the A1, A2, Table 1, and Table 2 data? Should mention references if external data. Should briefly describe how the data is obtained if authors' data.

Looking solely at Table 3, if we sum all revenues from electricity, gas, and carbon, Case 3 produces more revenues than Case 4. And, Case 4 is not more profitable than Case 1. Then what is the reason for using smart contracts if not more profitable? Even worse, why do we even need to consider P2G if we cannot profit more?

Why does this happen? Case 4 should generate more energies or power (Table 4), but why is it not more profitable?

Table 5 shows the performance of contract. Is this the smart contract or contract? If contracts, then what is a contract? If smart contracts, Case 1,2,3 do not consider smart contracts, why do we have these values?

Other possible minor fixes

  • Line 115, section 6 should be Section 6.
  • Figure 2, Electrolysis requires water, then why not include water as input in the diagram of electrolysis?
  • Line 240, "...a typical wind power curve k0..." and Line 245, "...typical wind power curve k..." How are they different? I think the authors need to rephrase their definitions.
  • Figure 3, in off-chain solving, "Smart contract uploads parameter information to off-chain section." I think the authors need to rephrase this sentence. This sentence indicates that the smart contract performs an upload to off-chain channels, which cannot be done in the smart contract. The smart contract can only perform on-chain operations, not off-chain.

Author Response

Dear reviewer,

Thanks a lot for your significant and precious comments on the manuscript. The authors take these comments seriously and have carefully revised related contents on the basis. Descriptions of the relevant responses are as follows.

Comment 1

Regarding the use of blockchain in the paper

Comment 1.1

First of all, the authors propose an off-chain solution as a workaround for complex on-chain operations. However, how do the authors guarantee the security and fairness of this off-chain solution?

Response 1.1

Many thanks to the reviewer for pointing out this issue, which was not clearly pointed out by the authors in the first draft. As added in the revised manuscript (lines 319 – 324, and lines 342- 347), the off-line solving part is the responsibility of the wind farm, and through the restrict of smart contract, dishonesty about how much energy the wind farm itself can provide will only result in penalties. This is what makes the application of smart contract in the scenario of multiple energy scheduling different from other application scenarios such as voting.

The authors are also particularly grateful to the reviewer for this summary sentence “propose an off-chain solution as a workaround for complex on-chain operations”, which the authors have used in the abstract (lines 15 – 16), and hope the reviewer will agree.

Comment 1.2

In lines 55-56, the authors said that "wind farms face the issue of insecure energy trading when participating in multiple energy markets. In addition, the electricity, gas, and carbon market are managed by different departments..." Considering this background, I cannot find solutions from the paper about the following issues.

  • Who should run the objective functions (Equation 11-14) as the off-chain solutions?
  • How can we trust that the person running those functions will always be honest and produce fair results?

Running the objective functions on-chain (as proposed by related works) can be considered secure and trustable because the smart contract execution is deterministic and hard-to-tamper due to the underlying consensus algorithm. However, if those functions are run off-chain, then how can you solve those questions?

Answering this question is crucial in a blockchain-based trading environment because participants are always untrusted and do not trust one another. Without answering this question, it is tough to pinpoint the usefulness of the blockchain in the paper. Mainly, how is the proposed system different if not using blockchain?

Response 1.2

Thanks a lot for this constructive comment. And the authors gratefully appreciate for your careful review and thoughtful comments.

In the modified Section 4 (lines 287 - 375), the authors strictly distinguish between the respective tasks of on-chain /off-chain, and the corresponding participators. A quick glance can also be found in modified Figure 3 (line 304). In brief, the wind farm should run the objective function (Equation 11-14) as an off-chain task and obtain a set of potential scheduling results based on multiple predictions of wind power output curves; then, the wind farm owner and the multiple energy market organizers run an on-chain smart contract, i.e, the Open Vote Protocol, to determine one particular result to agree on. Under this framework, the on-chain smart contract can be still considered as secure and trustable. Comparisons on case 4 against case 3 are mainly about the adoption of using blockchain-enabled smart contract, and descriptions are also modified in lines 394. It is hoped that the improved descriptions are clearer and more accurate than the original draft.

Comment 1.3

The contract items and contract status from Figure 1 are not explained. In addition, from my understanding, the contract items from Ref [25] are different meanings from the smart contract definition in general (e.g., the one in Ethereum). They are not the same.

Line 330-331, "It is worth noting that the performance of smart contract refers to the extent to which the scheduling plans of wind power curve k0 are fulfilled." This sentence is unclear. I think the authors need to describe better the "performance of smart contract" because it is used in their evaluations.

Usually, the "performance of a smart contract" is measured through a gas-used parameter, which as the authors mentioned, the simpler the better because of GasLimit. However, the authors define their own version of "performance of smart contract", which I cannot understand in this current manuscript due to lack of description.

Response 1.3

Thanks a lot for these valuable comments. And the authors sincerely apologize for the confused usages about contract, smart contract, contract items, and performance of smart contract. Many related expressions have been modified in the revised manuscript and are explained as follows.

Firstly, Figure 1 (line 163) is reorganized, and descriptions about it (lines 124 - 161 ) has been written as well. And many thanks for the reviewer to pointing out that the exact meaning of contract items from Ref [25]. This greatly helped the authors to better understand Ref [25] and enhance relevant phrases in this manuscript.

Secondly, the whole expression of "performance of a smart contract" is replaced by “the performance of adopting smart contract”, and details will expanded in Response 2.3. The authors do not propose a new parameter like GasLimit. The authors apologize again for these confusing expressions and hope that relevant improvements will be effective.

Comment 2:

Regarding implementation and evaluation

Comment 2.1

Where do the authors get the A1, A2, Table 1, and Table 2 data? Should mention references if external data. Should briefly describe how the data is obtained if authors' data.

Response 2.1

Thank you very much for this important reminder. The authors did not list sources of these parameters. In the revised manuscript, these references which are adopted for parameter settings have been listed. Data for Figure A1 and Figure A2 are from [34] and [35] (data processing details are given in lines 380-383). Data for Table 1 and Table 2 are mainly obtained from [37] and [30–31] with some assumptions (lines 397 – 399). And the references have been added as well.

Comment 2.2

Looking solely at Table 3, if we sum all revenues from electricity, gas, and carbon, Case 3 produces more revenues than Case 4. And, Case 4 is not more profitable than Case 1. Then what is the reason for using smart contracts if not more profitable? Even worse, why do we even need to consider P2G if we cannot profit more?

Why does this happen? Case 4 should generate more energies or power (Table 4), but why is it not more profitable?

Response 2.2

Thank you very much for pointing out this important problem. The authors gratefully appreciate for the reviewer’s careful review and profound comments. As a result of the review, the authors have further adjusted the explanations focusing on Table 3. In addition, the valuable comments of the reviewers have motivated the authors to further check the details, e.g., whether the setting of the penalty value affects the economic performance, and to add comparisons. The details are as follows.

The first modification focuses on the original comparison in Table 3 (line 434) and Table 4 (line 451). In the original manuscript, the authors’ idea was that, if the merits of the different cases were compared, it is designed to compare the total net revenue Y taking into account the cost of wind curtailment and the penalty of violating the contract items in smart contract, rather than the sum of the revenues from different energy markets. However, the reviewer's comments sufficiently indicate that this design by the authors is not thoughtful enough, so, the authors adjust the penalty values (lines 576 - 579) to make sure a better performance of case 4 no matter how it is interpreted. However, during this first modification, the authors found the importance of the setting of these values when adjusting them.

Thus, the second modification is to illustrate the influence of setting the values of penalties. Added simulations under different penalties in case 3 and case 4 are shown in Table 6 (line 480), and are analyzed. it is further observed that when this penalty for violating contract items is small, the effect of considering the performance of adopting smart contract during scheduling on Y is not significant, or even harms its effectiveness. In practical uses, the values of penalized deposits in the smart contract can be selected by experience and more tests (lines 471 – 480).

This comment from reviewer has greatly improved the quality of case study. The authors again express their gratitude.

Comment 2.3

Table 5 shows the performance of contract. Is this the smart contract or contract? If contracts, then what is a contract? If smart contracts, Case 1,2,3 do not consider smart contracts, why do we have these values?

Response 2.3

Thanks for the reviewer’s valuable comment. And the authors sincerely apologize for the confusion caused by “the performance of contract”. Firstly, as also pointed out by the reviewer in Comment 1, the contract here means smart contract. Secondly, subsection 5.2.3 is designed to compare is the impact of whether or not to adopt the smart contracts proposed in this paper on the results of the automated scheduling framework. Thus, the corresponding descriptions are modified to “the performance of adopting smart contract”.

As to the results, Cases 1, 2, and 3 do not consider the performance of contract in their objective functions, but the violation of energy volume according to their scheduling results can be calculated. Then, violated volumes and corresponding penalties can be obtained along with the total net revenue.

Comment 3

Other possible minor fixes

(1) Line 115, section 6 should be Section 6.

(2) Figure 2, Electrolysis requires water, then why not include water as input in the diagram of electrolysis?

(3) Line 240, "...a typical wind power curve k0..." and Line 245, "...typical wind power curve k..." How are they different? I think the authors need to rephrase their definitions.

(4) Figure 3, in off-chain solving, "Smart contract uploads parameter information to off-chain section." I think the authors need to rephrase this sentence. This sentence indicates that the smart contract performs an upload to off-chain channels, which cannot be done in the smart contract. The smart contract can only perform on-chain operations, not off-chain.

Response 3

The precision and patience of reviewer is greatly appreciated by the authors.

(1) The “section 6” has been corrected to “Section 6”, and is in Line 122 in the revised manuscript.

(2) Water has been added in Figure 2.

(3)The definition of typical wind power curves has been rephrased accordingly.

(4) Figure 3 and all related descriptions in Section 4 (lines 287 -375) have been improved. Indeed, as the reviewer points out, the off-chain solving and the on-chain smart contract are used to handle different tasks in the automated scheduling, and the smart contract can only perform on-chain operations, not off-chain. The authors' original presentation of the processes was not rigorous enough and has been corrected.

Once again, the authors express their sincere thanks to the reviewer. Any improvements are credited to the reviewer.

Round 2

Reviewer 3 Report

The authors have addressed my concerns well in the revised manuscript.